# Where Eriophyoidea (Acariformes) Belong in the Tree of Life

**DOI:** 10.3390/insects14060527

**Published:** 2023-06-06

**Authors:** Samuel J. Bolton, Philipp E. Chetverikov, Ronald Ochoa, Pavel B. Klimov

**Affiliations:** 1Florida State Collection of Arthropods, Division of Plant Industry, Florida Department of Agriculture and Consumer Services, Gainesville, FL 32608, USA; 2Zoological Institute of Russian Academy of Sciences, Universitetskaya nab., 1, St. Petersburg 199034, Russia; pchetverikov@zin.ru; 3Agricultural Research Service, Systematic Entomology Laboratory, United States Department of Agriculture, Beltsville, MD 20705, USA; ron.ochoa@usda.gov; 4Department of Biological Sciences, Purdue University, West Lafayette, IN 47907, USA; pavelklimov@yahoo.com; 5Institute of Environmental and Agricultural Biology (X-BIO), University of Tyumen, Tyumen 625003, Russia

**Keywords:** Acariformes, Eriophyoidea, Nematalycidae, Phylogenomics, Trombidiformes

## Abstract

**Simple Summary:**

Eriophyoidea (gall mites) are a group of four-legged, vermiform plant feeders that are among the most economically damaging groups of mites. Gall mites (although not all of them cause galls) were thought to belong within Trombidiformes, which represents a very large group of mites that includes other plant-feeding taxa, such as spider mites (Tetranychidae) and flat mites (Tenuipalpidae). Most recent phylogenetic analyses show no support for this taxonomic assignment. Instead, gall mites are almost certain to be closely affiliated with Nematalycidae, a group of vermiform mites that exclusively live in the soil. The small number of analyses that support the placement of gall mites within Trombidiformes are compromised by a lack of data, being based on only a few genes and/or excluding critical taxa. The majority of analyses support the placement of gall mites outside Trombidiformes.

**Abstract:**

Over the past century and a half, the taxonomic placement of Eriophyoidea has been in flux. For much of this period, this group has been treated as a subtaxon within Trombidiformes. However, the vast majority of recent phylogenetic analyses, including almost all phylogenomic analyses, place this group outside Trombidiformes. The few studies that still place Eriophyoidea within Trombidiformes are likely to be biased by incomplete taxon/gene sampling, long branch attraction, the omission of RNA secondary structure in sequence alignment, and the inclusion of hypervariable expansion–contraction rRNA regions. Based on the agreement among a number of independent analyses that use a range of different datasets (morphology; multiple genes; mitochondrial/whole genomes), Eriophyoidea are almost certain to be closely related to Nematalycidae, a family of vermiform mites within Endeostigmata, a basal acariform grade. Much of the morphological evidence in support of this relationship was apparent after the discovery of Nematalycidae in the middle of the 20th century. However, this evidence has largely been disregarded until very recently, perhaps because of overconfidence in the placement of Eriophyoidea within Trombidiformes. Here, we briefly review and identify a number of biases, both molecular- and morphology-based, that can lead to erroneous reconstructions of the position of Eriophyoidea in the tree of life.

## 1. Introduction

Eriophyoidea are a highly diverse lineage of plant-feeding arthropods that have proven to be elusive with respect to their placement within Acariformes [1]. Throughout the history of acarology, this group has been shifted around to very different taxonomic positions, including both within and outside Trombidiformes. From the mid-twentieth century until very recently, it was widely thought that Eriophyoidea belong within Trombidiformes, but in the past few years this hypothesis has been greatly undermined by a series of molecular phylogenetic analyses [2,3,4,5,6,7,8,9,10] and a new morphology-based analysis [11]. However, several recent phylogenetic analyses have recovered Eriophyoidea within Trombidiformes [12,13,14]. A review is therefore needed to gauge the level of uncertainty that remains with respect to the phylogenetic position of Eriophyoidea. In this paper, we attempt to explain incongruences among the results of all phylogenetic analyses that bear on this topic. Our review begins by tackling historical (pre-molecular) hypotheses on the position of Eriophyoidea, which helps to explain the origin and persistence of errors and misinterpretations that still affect this topic. The review then addresses the results of various molecular-based analyses before ending with a rebuttal to a recently published phylogenetic analysis that reports the recovery of Eriophyoidea as nested within Tydeoidea [12].

## 2. The Morphological Era (1877–2015)

Two very early classification schemes for mites, which both date to 1877 [15,16], placed Eriophyoidea within Acaridae (=Astigmata). This was due, at least in part, to the absence of stigmata. Shortly afterward, in 1884, Claus and Moquin-Tandon [17] treated Eriophyoidea as a separate and distinct group from all the other main branches of mites recognized at the time. Eriophyoids have also been grouped with Demodicidae in Vermiformia based on their shared worm-like body form [18], but this placement was not followed in the classification schemes of Oudemans [19] and Reuter [20]. Due to a shared feeding ecology, Oudemans suspected that Eriophyoidea were closely affiliated with phytophagous taxa within Trombidiformes. However, Reuter rejected this idea based on the morphology of the digestive system. Trombidiformes and Oribatida have a diverticulum (caecum) comprising a pair of cavities that branch away from the main passage of the gut [21]. Reuter [20] argued, probably correctly, that the absence of a diverticulum in Eriophyoidea was reason to doubt a close relationship between Eriophyoidea and Trombidiformes. In recent decades, the morphology of the digestive tract has not been considered with respect to basal relationships among mites. Therefore, a re-examination of this system might reveal highly conserved characters that are congruent with the results of recent phylogenomic analyses.

Baker and Wharton [22] treated Eriophyoidea as one of three basal lineages within Trombidiformes, the others being Tarsonemini and Prostigmata. Somewhat confusingly, they also inferred that eriophyoids share a recent common ancestry with Tetranychidae and Phytoptipalpidae (=Tenuipalpidae), which are in a relatively derived position within Trombidiformes [4,5]. They based this inference on the following characters: stylet-like chelicerae, phytophagy, rayed empodia (Eriophyoidea and Tenuipalpidae), and elongate bodies (Eriophyoidea and some Tenuipalpidae) [22,23]. For several decades, eriophyoids were widely accepted to be affiliated with Tetranychoidea [21,24,25]. However, this position is incongruent with a number of plesiomorphies present in eriophyoids, namely, indirect sperm transfer, the possession of fundamental setae on coxisterna II, and the presence of eugenital setae in males [26]. Therefore, this group was relocated to a more basal position, specifically as sister to Tydeoidea [26,27,28]. This new placement was largely based on the aforementioned plesiomorphies and a suite of homoplastic paedomorphisms (see below).

It is remarkable that a close relationship between Eriophyoidea and Nematalycidae was never proposed as likely until very recently [11]. These taxa share a number of similarities that are readily apparent without the need for a detailed morphological analysis, including an annulated and worm-shaped body, unpaired *vi* (the rostral seta on the prodorsum) when present, and the absence of prodorsal trichobothria and body lyrifissures (cupules). Keifer [29] noted that both taxa have genitalia that are relatively anteriorly positioned on a worm-like body. However, he used these similarities, probably assumed by him to be convergent, only to explain the loss of legs III and IV in Eriophyoidea rather than infer a close phylogenetic relationship. Shevchenko et al. [30] did not rule out the possibility of a close phylogenetic relationship between Eriophyoidea and Nematalycidae, but they hypothesized that similarities between these two taxa were more likely to be due to convergent evolution. 

In the period leading up to the relocation of Eriophyoidea to a more basal position within Trombidiformes, the position of Nematalycidae was also in flux. Whereas Nematalycidae were originally thought to belong in Endeostigmata [31], Cunliffe [32] hypothesized that this family is more closely related to Tydeoidea, but he provided no arguments to support this hypothesis. In accordance with Cunliffe, in the following decades, most authors placed this family within Tydeoidea [33,34] or as a separate superfamily that is allied to Tydeoidea [35]. Nematalycidae and Tydeoidea share a number of characters that are either plesiomorphic (indirect sperm transfer and the retention of fundamental setae on coxisterna II) or homoplastic (undivided femora, fusion of the palpal femur with the palpal genu). Other characters strongly suggest that Nematalycidae fall outside Trombidiformes. For example, nematalycids clearly bear rutella [36,37,38], structures that are widely thought to be lost throughout Trombidiformes [36,39]. Eriophyoids bear infracapitular guides, which are possible homologues of rutella [38,40]. Additionally, unlike Tydeoidea, both Eriophyoidea and Nematalycidae lack stigmata on the proterosoma or anywhere else on the body, weakening support for their placement within Trombidiformes.

Although Kethley treated Nematalycidae as allied to Tydeoidea [35], soon afterwards he hypothesized that Nematalycidae group with Micropsammidae and Proteonematalycidae within Nematalycoidea [41]. However, in his discussion of the support for this relationship, the only character he listed that is shared by all three families is the absence of trichobothria. He made no mention of where Eriophyoidea belonged, and there is no evidence that he considered the possibility that Eriophyoidea is affiliated with Nematalycidae. 

Lindquist clearly did consider the possibility of a close relationship between Eriophyoidea and Nematalycidae, but he largely rejected this idea in favor of a sister relationship between Eriophyoidea and Tydeoidea [27]. One of his main reasons for doing so was that Nematalycoidea, which is almost certain to be an artificial group [4,11], was hypothesized to be in a basal position, outside of Trombidiformes (largely because of the shared possession of rutella). By this time, Lindquist had also undertaken an unpublished cladistic analysis of Trombidiformes, in which Eriophyoidea was recovered as sister to Tydeoidea (Lindquist, pers. comm. 10 April 2023). A single cladogram from this analysis was published without any associated data [28]. Lindquist excluded Nematalycidae from the analysis because, by this time, Nematalycoidea had already been erected by Kethley [41]. This omission could have caused Eriophyoidea to group with Tydeoidea as a result of a suite of shared apomorphies that probably represent homoplastic paedomorphisms (e.g., the partial or complete suppression of anamorphosis and the loss of urstigmata). The publication of a very similar phylogeny, which also lacks any associated data, suggests that Kethley arrived at the same result [26], but this was probably also because of the same omission of Nematalycidae from an analysis concerning only those taxa assumed to belong in Trombidiformes. 

Therefore, although Nematalycidae were relocated outside Trombidiformes, the same treatment was apparently not considered for Eriophyoidea. This was somewhat remedied by Lindquist when he suggested that Eriophyoidea may be closely affiliated with Alycidae [40]. However, later classification schemes followed Lindquist’s earlier work [39,42]. 

Lindquist contended that the consensus of morphological evidence favored a sister relationship between Eriophyoidea and Tydeoidea [27], although many of the characters that he used in support of this relationship show an equal or greater resemblance between Eriophyoidea and Nematalycidae [11]. For example, the ambulacra of Eriophyoidea bear a far closer resemblance to those of *Gordialycus*, a taxon within Nematalycidae that was recovered as sister to Eriophyoidea via a morphology-based phylogenetic analysis [11]. The empodia of *Gordialycus* and Eriophyoidea are very similar (Figure 1). Both are slender and weakly tapering structures that bear a small number of setules (usually less than eight pairs). In the case of *Gordialycus*, the setules (rays) are trifucated (Figure 1c), whereas in Eriophyoidea, they are typically either bifurcated or trifurcated (Figure 1b). By comparison, the pad-like empodia of Tydeoidea (Figure 1a) bear no real resemblance to the empodia of Eriophyoidea or Nematalycidae. However, several eriophyoid taxa, e.g., *Aberoptus* and *Cisaberoptus*, have aberrantly widened empodia that bear a superficial resemblance to those of some tydeoids ([43]: Figure 4E). But these eriophyoids have retained distinctly pointed empodia, whereas tydeoids have rounded empodia.

Of the characters used by Lindquist [27] to argue for a sister relationship between Eriophyoidea and Tydeoidea, the only ones that show a greater degree of resemblance between eriophyoids and tydeoids than between eriophyoids and nematalycids are a suite of strongly interdependent and homoplastic characters pertaining to paedomorphisms, including the partial or complete suppression of anamorphosis, the loss of urstigmata, and the suppression of genital papillae and the nymphal progenital chamber [11]. Note that the nymphal progenital chamber may even be present in basal Eriophyoidea; in the nymphs and larvae of *Pentasetacus araucariae*, a fovea between setae *3a* represents the nymphal progenital chamber [46]. 

## 3. Molecular Era (2016–Present)

The first molecular phylogenetic analysis to address the position of Eriophyoidea was based on a mitogenomic analysis and showed strong support for the placement of Eriophyoidea outside of Trombidiformes [2]. Subsequent mitogenomic analyses also recovered that result [6,7,8,9]. Only a single mitogenomic analysis, which is undermined by the undersampling of non-trombidiform taxa, has recovered Eriophyoidea from within Trombidiformes [13]. Whole-genomic analyses have also recovered Eriophyoidea from outside of Trombidiformes [3,10]. Therefore, there is near unanimity among whole-genomic and mitogenomic analyses. Conversely, phylogenetic analyses based on Sanger sequencing have produced no strong consensus on the approximate position of Eriophyoidea. In some cases eriophyoids were recovered from outside of Trombidiformes and near the base of Acariformes [4,5], whereas in others eriophyoids were recovered from within Trombidiformes [5,12,14]. Klimov et al. [5] recovered Eriophyoidea within Trombidiformes only when the nuclear protein partition was used (EF1-α, SRP54, HSP70), whereas Eriophyoidea was recovered outside of Trombidiformes when either all six loci (COI, 18S rRNA, 28S rRNA, EF1-α, SRP54, HSP70) or only rDNA loci (18S rRNA, 28S rRNA) were used. 

A substantial proportion of phylogenetic analyses provided support for a close relationship between Eriophyoidea and Nematalycidae; Eriophyoidea are either sister to Nematalycidae [4,5] or nested within Nematalycidae [4,11]. These analyses are based on morphology [11], Sanger sequencing [4,5], and whole genomes [3]. The whole genome analysis, which is the only phylogenomic analysis to date to have included endeostigmatic taxa, revealed that Eriophyoidea and Nematalycidae share a suite of highly conserved nuclear proteins [3]. With respect to all the phylogenetic analyses that recovered Eriophyoidea outside of Trombidiformes, their congruence with a close relationship between Eriophyoidea and Nematalycidae is strong support for this relationship and also for the placement of Eriophyoidea outside of Trombidiformes (Figure 2). 

On the other hand, there is very little consistency in the results of analyses that have recovered Eriophyoidea from within Trombidiformes. Eriophyoidea have been found in the following positions: (1) sister to a clade comprising Tydeoidea, Eupodidae, and Adamystidae [5: 18S rRNA, 28S rRNA, EF1-α, SRP54, HSP70, and COI]; (2) sister to a clade comprising Parasitengona, Cheyletoidea, and Eupodides [13: mitochondrial genomes]; (3) nested within Tydeoidea [12: 18S rRNA, 28S rRNA, and COI]; (4) sister to Tydeoidea [14: 12S rRNA, 18S rRNA, 28S rRNA, and COI]; (5) sister to a clade comprising Nematalycidae and *Benoinyssus* (a genus within Eupodidae) [14: 12S rRNA, 18S rRNA, 28S rRNA, and COI]. The incongruences among these results undermine the case for the placement of Eriophyoidea within Trombidiformes. In all of these analyses, Eriophyoidea have probably been pulled into the basal region of Trombidiformes as an effect of long-branch attraction [5], and in most cases this artifact was likely exacerbated by either the undersampling of basal taxa [13,14] or by the inclusion of hypervariable rDNA regions [12], which are unalignable by their nature (see below).

## 4. Are Eriophyoidea Nested within Tydeoidea?

More detailed consideration will now be given to the results of Szudarek-Trepto et al. [12] (from here on abbreviated to ST22) because they recently recovered Eriophyoidea as nested within Tydeoidea, which is a similar result to the sister relationship that was hypothesized for the two taxa [26,27,28]. The molecular phylogeny of ST22 was based on only three loci (18S rRNA, 28S rRNA, and COI), and the results are undermined by a failure to account for the rRNA secondary structure (rDNA stem complementarity) as the basis of alignment. Moreover, ST22 undertook a manual alignment for genera and families without testing for their monophyly first, which represents a form of circular reasoning (an alignment favoring traditional groupings can skew an analysis toward a traditional phylogeny). The authors also state that inconsistencies with the results of other analyses can be partly explained by a “different alignment strategy without the removal of regions that are difficult to align”, thus advocating for the use of hypervariable regions in deep phylogenetic reconstructions. 

Hypervariable regions experience frequent expansion–contraction and instability in corresponding stem-loop rRNA structures, so sequence homology cannot be established here, except for closely related taxa [47,48,49]. This major methodological flaw further undermines the results of ST22. When attempting to undertake a phylogenetic analysis to resolve internal relationships within Phytoptidae (Eriophyoidea), Chetverikov et al. [50] had to exclude the hypervariable regions because they were unalignable. We consider the major inconsistencies of the hypervariable regions at the intrafamilial level to be one of the strongest arguments against the acceptance of the results of ST22, which pertains to relationships at the level of superfamily and suborder. When the secondary structure of 18S and 28S rRNA is accounted for, and hypervariable expansion–contraction regions are removed, eriophyoids are placed in a close affiliation with nematalycids, as indicated by a recent study [4]. 

ST22 also recovered Eriophyoidea nested within Tydeoidea based on a separate morphology-based analysis. Unfortunately, they omitted Nematalycidae (and any other Endeostigmata), thereby disregarding alternative hypotheses. Only four superfamilies were included in the ingroup, thus greatly constraining the morphological analysis to produce the same result as the molecular analysis. ST22 claim that eight characters “unambiguously define” an Eriophyoidea-Tydeoidea clade. However, only two of these characters are synapomorphies (characters #1, #2), and the remaining six are homoplasies. Moreover, the form of the movable digit (character #1) was not coded for Eriophyoidea, so only a single synapomorphy remains for this group, chelicerae fused at bases (character #2.1). However, the evidence for this character state in Eriophyoidea is extremely weak [11]; in an early derivative eriophyoid taxon, *Pentasetacus plicatus*, the cheliceral bases appear unfused ([51]: Figure 5D,E). Therefore, the proposed Eriophyoidea-Tydeoidea grouping is not supported by a single synapomorphic character state.

There are also other apparent weaknesses in the coding. The third character (#3) addresses the potential absence of cheliceral setae *chb* in Eriophyoidea and Tydeoidea. However, the absence of *chb* in Eriophyoidea cannot be asserted with confidence ([11,51] (Figure 5D,E) and [52]). Secondly, in both Eriophyoidea and Tydeoidea, setae *ps3* (the third pair of setae on the pseudanal (*PS*) segment) were coded as absent (character #41), but in Eriophyoidea the absence of *ps3* is attributable to the hypothesized absence of the pseudanal segment, whereas the absence of *ps3* in Tydeidae is not. Setae *ps3* should have therefore been coded as unknown or “not applicable” in Eriophyoidea. The miscoding of this character generated the only morphological support for the nested position of Eriophyoidea within Tydeoidea. 

With the single exception of the absence of urstigmata, the homoplasious character states used by ST22 to support an Eriophyoidea-Tydeoidea clade (absence of naso; absence of urstigmata; undivided femur I; undivided femur II; undivided femur III, but note that this is inapplicable to Eriophyoidea; absence of rhagidial organs) are also shared with Nematalycidae [11]. On the other hand, Nematalycidae share at least eight character states with Eriophyoidea that are not shared with Tydeoidea, including the following: annulated body; unpaired *vi* (although most eriophyoids and some nematalycids lack this seta); slender empodia with bifurcating or trifurcating setules; legs II lacking lateral claws (Eriophyoidea and derived Nematalycidae); absence of palp tarsal solenidion; absence of stigmata; absence of prodorsal trichobothria; absence of cupules. Unfortunately, the phylogenetic value of these character states was not evaluated by ST22 due to both character and taxon omission.

Of all the different and competing hypotheses on the position of Eriophyoidea, a nested placement within Tydeoidea must be one of the least likely because it is incongruent with the fundamentally different modes of gnathosomal integration of these two taxa. Gnathosomal integration proceeded in Tydeoidea via the reduction of the fixed digits, so that a cheliceral groove formed along the dorsum of the subcapitulum around the movable digits [53]. This reduction circumvented the need for the modification of any part of the subcapitulum into a sheath, as instead occurred in Eriophyoidea and some Nematalycidae [38]. Therefore, if Eriophyoidea is nested within Tydeoidea, the complex mode of gnathosomal modification that arose in the latter would have to have been undone, including the re-lengthening of the fixed digits, before an altogether different and no less complex mode of gnathosomal integration could proceed, namely the envelopment of the chelicerae within a subcapitular sheath. Thus, a nested position within Tydeoidea is not at all parsimonious with respect to morphology.

## 5. Conclusions

Almost all of the molecular phylogenetic analyses that have recovered Eriophyoidea within Trombidiformes have involved questionable methodologies, namely the undersampling of basal taxa [13,14] or the inclusion of hypervariable rDNA regions [12]. The only other analysis to recover Eriophyoidea from within Trombidiformes was based on only three loci [5] (the same study recovered Eriophyoidea as outside Tromidiformes when six loci were used). All other recent phylogenetic analyses are congruent with the placement of Eriophyoidea outside of Trombidiformes and in a close relationship with Nematalycidae. Were it not for the once prevailing assumption that Eriophyoidea belongs within Trombidiformes, this relationship would have likely been strongly apparent before the molecular era from a suite of distinctive and readily discernible morphological characters. The combination of three recent phylogenetic analyses have more or less corroborated a close relationship between Eriophyoidea and Nematalycidae because each analysis is based on a different dataset: morphology, whole genomes, and Sanger sequencing [3,4,11]. However, it is not yet clear if Eriophyoidea is sister to Nematalycidae or nested within Nematalycidae. The morphological evidence favors a nested relationship [11], although this is weakly supported. The molecular evidence is ambiguous [4]. Phylogenomic analyses may soon resolve this question.

## Figures and Tables

**Figure 1 insects-14-00527-f001:**
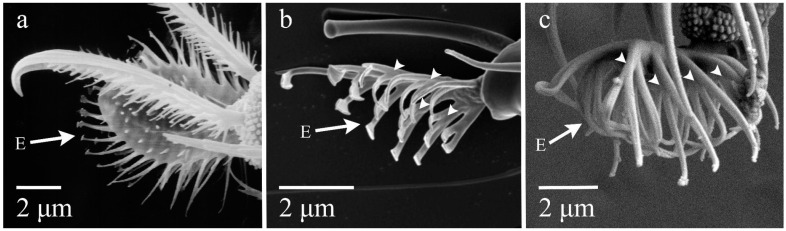
Empodia. (**a**) *Tydeus* sp. (Tydeoidea); (**b**) *Aceria* sp. (Eriophyoidea); (**c**) *Gordialycus* sp. (Nematalycidae). E = empodium. Arrowheads point to trifurcations near the base of each setule. These images were generated using the methods detailed in Bolton et al. [44,45].

**Figure 2 insects-14-00527-f002:**
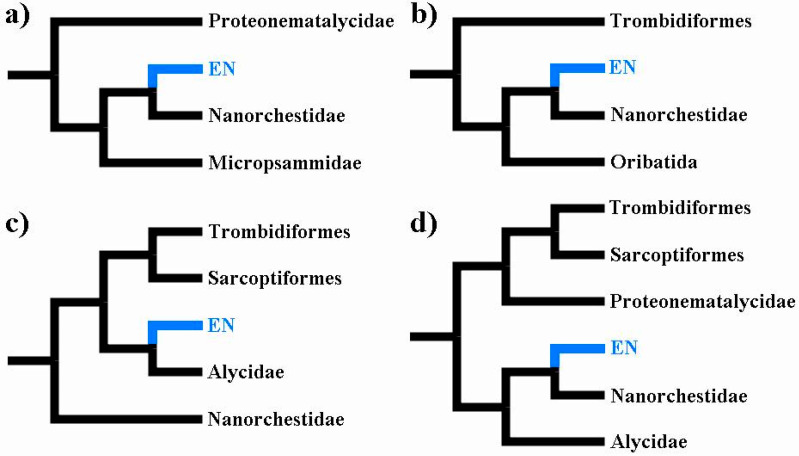
Summary trees pertaining to analyses that have recovered an Eriophyoidea-Nematalycidae (EN) clade (highlighted blue): (**a**) parsimony tree (showing only part of Sarcoptiformes) based on 110 morphological characters [11]; (**b**) maximum likelihood tree (low taxonomic resolution) based on 90 orthologous proteins [3]; (**c**) maximum likelihood tree based on COI, 18S rRNA, 28S rRNA, EF1-α, SRP54, and HSP70; [5]; (**d**) Bayesian time tree based on COI, 18S rRNA, 28S rRNA, and HSP70 [4]. Across all trees, Nematalycidae were recovered as paraphyletic (**a**) or sister (**c**,**d**) to Eriophyoidea—taxonomic sampling was insufficient to resolve this for the 90 orthologous proteins (whole genome data) (**b**).

## Data Availability

The data presented in this study are available in this article.

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
