# Peer review of "Where Eriophyoidea (Acariformes) Belong in the Tree of Life"

_insects, 2023, doi:10.3390/insects14060527_

Round 1
Reviewer 1 Report
This is a brief review on an important, and rather controversial topic: the phylogenetic position of gall and rust mites (Eriophyoidea) within Acari. It is written well and is generally fair in its argumentation. Besides a few possible adjustments and ‘tightening’ of the text (in particular the conclusion), I find it quite acceptable for publication. I would suggest to be give extra care in the way that the conclusion is written, to be slightly more ‘neutral’. Having said that, this manuscript is useful as a whole, to emphasize on the most likely hypotheses if relationships.
Conclusion:
-I suggest to slightly adjust the conclusion so that it’s written in a more objective way. For instance: since virtually all analyses have incomplete sampling of both characters and taxa, saying that this is ‘questionable’ make it sound like all analyses are questionable, which was not your point. I do understand, however, why you used the word questionable, and perhaps you can still use it, after having adjusted the sentence.
-Line 297: you say that “main results of all analyses are congruent…”: “all” seems to be an exaggeration, since analyses include both morphological or DNA analyses, and regardless how flawed they may be, not ‘all’ analyses were congruent with such placement. I suggest you adjust moderately the wording.

Author Response
This was extraordinarily useful feedback. Thank you very much to the reviewer. I went with all recommendations except on a few occasions, see below.
Line 130. Yes, this reference is correct.
Line 196. I don't believe there is redundance here. Consider the inverse case. If Eriophyoidea was consistently found with Tydeoidea when recovered within Trombidiformes while also being found in a range of different positions throughout Endeostigmata, would that not greatly undermine the case for the placement of Eriophyoidea outside Trombidiformes? The similarities among analyses in the recovery of the specific location is also strong support for the general location.
Line 267. Those other characters support an eriophyoidea-nematalycidae clade, but only a single character supports a nested position of Eriophyoidea within Tydeoidea (a clade comprising Tydeidae, Ereynetidae and Eriophyoidea but not Iolinidae). It is not among the 8 characters they use to support a clade comprising all Tydeoidea and Eriophyoidea.
Line 282. A sister relationship between Eriophyoidea and Nematalycidae (as Lindquist argued) is actually not that objectionable from the point of view of the mouthparts. ST22 make the case that the recovery of a nested relationship based on both their morphological and molecular analysis is strong support for the nested relationship. But the nested relationship is greatly undermined by the mouthparts. This is the point of that specific paragraph. Note the title of the section (Are Eriophyoidea nested within Tydeoidea?). The general weaknesses of the methodology were covered early in the section, but I would argue that the greatest problem with their chief result is that it makes no sense from the point of view of the mouthparts.
Line 293. This last sentence only pertains to the gnathosoma as evidence against the nested relationship, so 'thus' is preferred over 'overall'.

Reviewer 2 Report
The paper summarizes and gives careful details on the topic. I do not have any comments for the text. It is a very specialistic paper in Acarology and its information and evaluation are well supported.
Author Response
Thanks for the positive feedback. Note to editors: no more is added here because no amendments were suggested.
Reviewer 3 Report
It was a pleasure for me to read this short but interesting review, which sheds light on the taxonomic classification of one particular group of mites. The review is well written and, in addition to its scientific contribution to mite taxonomic community, it also provides an interesting example of good practice in taxonomy and in biology in general - one cannot rely only on historical assumptions, single studies and only one type of data. This study emphasizes the complex approach, encourages the avoidance of exaggerated conclusions and thus provides a great take home message also for wider audience. I have just few minor comments:
Line 13: “Summary” should be in bold
Line 92 and 274: what does the abbreviation “unpaired vi” exactly mean? It may be little confusing for the people who are not familiar with acarologist's jargon like me. Similarly, please add further explanation (one or two words) what “PS” and “ps3” mean (Lines 263-265).
Line 152: please correct typo “trifucated”
Author Response
Thanks very much to the reviewer for the feedback.
Line 13 appears to be a problem with the PDF generator of the MDPI website. In the original text Simple Summary is all bold.
Line 92. setae vi now also described as "the rostral seta on the prodorsum".
Lines 263-265. ps3 is now explicitly defined as the third pair of setae on the pseudanal (PS) segment. This hopefully now clarifies these terms.
